# Effectiveness of a Parent Training Programme for Parents of Adolescents with Autism Spectrum Disorders: Aiming to Improve Daily Living Skills

**DOI:** 10.3390/ijerph19042363

**Published:** 2022-02-18

**Authors:** Nanako Matsumura, Haruo Fujino, Tomoka Yamamoto, Yuki Tanida, Atsuko Ishii, Aika Tatsumi, Mariko Nakanishi, Masaya Tachibana, Ikuko Mohri, Hiroko Okuno

**Affiliations:** 1Department of Child Development, United Graduate School of Child Development, Osaka University, Suita 565-0871, Osaka, Japan; u124832k@ecs.osaka-u.ac.jp (N.M.); ikuko@kokoro.med.osaka-u.ac.jp (I.M.); 2Molecular Research Centre for Children’s Mental Development, United Graduate School of Child Development, Osaka University, Suita 565-0871, Osaka, Japan; t-yamamoto@kokoro.med.osaka-u.ac.jp (T.Y.); aishii@kokoro.med.osaka-u.ac.jp (A.I.); aika-tatsumi@kokoro.med.osaka-u.ac.jp (A.T.); nakanishi@kokoro.med.osaka-u.ac.jp (M.N.); m-tachi@kokoro.med.osaka-u.ac.jp (M.T.); 3Japan Society for the Promotion of Science, Tokyo 102-0083, Japan; tanida.yuki@kokoro.med.osaka-u.ac.jp; 4Graduate School of Humanities and Sustainable System Sciences, Osaka Prefecture University, Sakai 599-8531, Osaka, Japan; 5Graduate School of Nursing of Health and Human Science, Osaka Prefecture University, Habikino 583-8555, Osaka, Japan; okuno.h@nursing.osakafu-u.ac.jp

**Keywords:** autism spectrum disorder, parent training, behaviour therapy, adolescents, daily living skills, adaptive behaviour

## Abstract

Parent training (PT) has been well established in younger children with autism spectrum disorder (ASD) but is less well studied in adolescents. This study examined the effects of attempting PT to enhance the daily living skills (DLSs) of adolescents with ASD. Twenty-five parents of adolescents with ASD participated in either the immediate- or delayed-treatment control condition. Children’s DLSs were evaluated using the DLS domain of the Vineland Adaptive Behaviour Scales-II, and the achievement of the DLSs practised by the children at home was the subject of the evaluation. The DLS domain score showed no improvement in the treatment group compared to the control group. However, some parents in the treatment group reported that their children acquired the target DLSs and more sophisticated communication behaviours. In addition, one measure suggested that parents increased their praising behaviours. These changes may have been driven by the completion of the parent training. We discuss several aspects of developing parent-mediated interventions based on the current intervention situation and observed changes.

## 1. Introduction

### 1.1. Difficulty in Acquiring Daily Living Skills (DLSs) for Those with Autism Spectrum Disorder (ASD)

Autism spectrum disorder (ASD) is a neurodevelopmental disorder characterised by impaired social communication and limited and repetitive patterns of behaviour, interests, or activities [1]. A critical problem for children with ASD is impairment in daily living skills (DLSs), which are adaptive behaviours for independent living, such as self-reliance (e.g., meal preparation, dressing, and hygiene management), housework (e.g., cleaning and washing), and community living (e.g., time and money management) [2]. Impairment in DLSs results in the need for extensive assistance in daily life in relation to social contacts, employment sites, and economic management [3]. In particular, impairments in DLSs can occur in adolescents with ASD, even if they have no intellectual disability [4,5,6]. Although various factors can be associated with impairment in DLSs, such as ASD characteristics, age, sex, intelligence quotient (IQ), and executive function [5,6,7,8,9,10], developing methods to facilitate the acquisition of DLSs in adolescents to support their ability to live independently is desirable.

An additional problem related to ASD is psychological stress associated with parenting. Parents of children with ASD experience higher stress in parenting than parents of typically developing children [11,12]. Such stress is at least partially rooted in DSL-related impairment. For example, studies have reported that parenting stress is related to poor functional independence in children with ASD [13], which requires more parenting behaviour and reduces parents’ leisure time [12]. This is also the case for adolescents with ASD because their various physiological and psychological changes make it more difficult for both children with ASD and their parents to communicate their parenting difficulties. Indeed, some parents of adolescents with ASD must manage their children’s daily living behaviours, such as home care, transportation, money management, self-care, and other skills, even after high school graduation [14]. Therefore, the facilitation of the acquisition of DLSs by children with ASD is desirable to reduce parental stress.

### 1.2. Parent Training (PT)

Parent training (PT) is a parent-mediated intervention based on behavioural theories that was developed to support parents of children with ASD [15]. Parents play the role of co-therapists in PT and are expected to change their parenting behaviours to increase adaptive behaviours in their children [16,17,18].

Cumulative evidence has suggested that PT for parents of children with ASD not only increases children’s adaptive behaviours (communication and daily living skills), but also reduces parental stress and improves parent–child relationships [15,19,20]. However, a recent systematic review suggested that some parent-mediated interventions for younger children (i.e., 10 or younger) are effective, but there is limited evidence of the effectiveness of PT in older children [21]. In Japan, one study attempted PT involving adolescents: Matsuo et al. [22] investigated the effects of a PT programme for parents of adolescents with ASD or other developmental disabilities and showed that improvements were limited to only the aspect of parent–child interactions. The potential effectiveness of PT in DLSs has not been examined in Japan.

This study conducted a DLS-specific PT with parents of adolescents with ASD, and examined its effectiveness based on the degree of DLS acquisition in their children. We examined changes in DLSs, interpersonal responsiveness, behavioural problems in children, parents’ mental health and self-confidence in responding to their child, and parent–child relationships.

## 2. Materials and Methods

### 2.1. Participants

Parents of adolescent children with ASD were recruited from the paediatric outpatient department of the Osaka University Hospital. The inclusion criteria were as follows: (1) the parents experienced difficulties in interacting with their children or had concerns about parenting; (2) their children were diagnosed with ASD; (3) the children were between 10 and 15 years old; and (4) IQs were between 65 and 99, as measured using the Japanese version of the Wechsler Intelligence Scale for Children-Fourth Edition (WISC-IV; [23]). Parents with any record of child abuse or other criminal offences, intellectual disability, or serious mental illness were excluded. Osaka University Hospital was a tertiary care institution (at the time of the study), and the degree of symptoms varied in patients. We consecutively recruited parents who met the inclusion criteria among those who visited the hospital during the recruitment period (August to November 2018). Recruitment was conducted as follows: when parents who met the criteria came to the hospital, their primary physicians or psychologists gave them a PT leaflet. Based on a previous PT study in Japan [22], we stopped recruiting once we achieved similar numbers in each group, and the sample size was 25 parents of children with ASD. In total, 25 families participated in this study. The children were aged 10 to 15 years (mean age = 12.07 years old, SD = 1.51). Experienced paediatricians diagnosed all children with ASD according to the diagnostic criteria of the Diagnostic and Statistical Manual of Mental Disorders Fifth Edition (DSM-5; [1]). Twenty parents had experience in receiving support for parenting or childcare (i.e., parent training and counselling, as well as applied behaviour analysis, occupational therapy, and speech therapy). This study was approved by the Ethics Review Committee of the Osaka University Hospital (no. 17454(TR17454)-3). Parents were informed of the content and purpose of the study using the research protocol and written consent form on which informed consent for the study was obtained.

The results of this study are shown in Figure 1. Participants were divided into an immediate-treatment (IT) group (*n* = 13) and delayed-treatment control (DTC) group (*n* = 12). The participants in the IT group completed an interview and self-administered questionnaires one month prior to the implementation of PT (Time 1) and within one month after the intervention (Time 2). In the DTC group, participants also completed an interview and self-administered questionnaires at the same time as those in the IT group (Time 1 and Time 2). After data collection at Time 2, the DTC group was offered the same PT programme content as the IT group.

### 2.2. Measurement

#### 2.2.1. Children’s Measurements

The children’s achievement of DLSs was evaluated by their parents between sessions 5 and 6 of the PT for each task set after parent–child discussions as part of the session 4 assignments. DLSs that were not performed by children were excluded. Those who received parental help were included in the study. Parents rated their child’s levels of acquisition of the DLSs on a 3-point scale: 1: independent; 2: likely to be independent (partially prompted or an increase in the behaviour); and 3: not achieved. Scores of 1 and 2 were considered to indicate that the skill was achieved.

This study used several conventional scales. The Japanese version of the Vineland Adaptive Behaviour Scales–Second Edition (VABS-II; [2,24]) was used to assess the children’s level of acquisition of adaptive behaviours. The VABS-II assesses children’s levels of various aspects, including communication, DLSs, socialisation, and overall adaptive behaviours. These assessments were performed as semi-structured interviews with a parent/caregiver familiar with the child’s condition. An interviewer scored each questionnaire item on a 3-point scale by marking a higher score for better adaptive function. Total scores were calculated for each of the following measures: communication, DLSs, socialisation, and adaptive behaviour.

Children’s psychosocial adjustment/maladaptive status was assessed using the Japanese version of the Child Behaviour Checklist (CBCL, [25,26]), which is composed of 113 items scored by a parent. It is applicable to ages 4 to 18 and consists of internalising and externalising scales with eight subscales: withdrawal, somatic complaints, anxiety/depression, social problems, thought problems, attention problems, aggression, and delinquent behaviour. Each item is rated on a 3-point scale. Higher scores indicate greater maladaptive behaviours. Internal consistencies of the scales in this study were adequate (Cronbach’s α = 0.68; internalising scale, α = 0.86 for externalising scale).

Children’s interpersonal responsiveness was measured using the Japanese version of the Social Responsiveness Scale, Second Edition (SRS-2; [27,28]), which is composed of 65 items and is applicable to 4- to 18-year-old children. This scale measures the ASD-related symptoms in daily life. Scoring for this scale was also performed by a caregiver who was familiar with the target child. It includes two subscales compatible with the DSM-5, namely social communication (SCI) and restricted interest and repetitive behaviours (RRB), and five clinical subscales, namely interpersonal awareness, interpersonal cognition, interpersonal communication, interpersonal motivation, and repetitive/restricted behaviour. Each item is rated on a 4-point scale. Higher scores indicated greater symptoms. Internal consistencies of the subscales in this study were acceptable to a high of 0.61–0.85, except for social awareness (Cronbach’s α = 0.18).

#### 2.2.2. Parents’ Measures

Parenting stress was assessed using the Japanese version of the Parenting Stress Index (PSI; [29,30]). This index consists of measures of parents’ stress, parent–child and family problems, and other factors. It yields the child and parent domain scores as well as the total score. The children’s mothers rated 78 items on a 5-point scale in terms of the severity of parenting stress. Higher scores indicated greater psychological stress. Internal consistencies of the subscales were high (Cronbach’s α = 0.81 for parent total and α = 0.85 for child total).

Parents’ mood was assessed using the Japanese version of the Beck Depression Inventory–Second Edition (BDI-II; [31,32]), assessing the presence and severity of depressive symptoms. Mothers rated 21 questions on a 4-point scale regarding their own condition during the previous two weeks. Higher scores indicate greater depressive symptoms. The internal consistency was high in this study (Cronbach’s α = 0.89).

Parenting behaviour was assessed using the Confidence Degree Questionnaire for families (CDQ; [16]), which measures the degree of parents’ confidence in managing their children. Although not standardised, it is used in Japan to measure changes in parents’ thoughts regarding parenting, responding to children, and parenting confidence before and after PT. The Japan Association of Parent Training recommends the CDQ for evaluating the effectiveness of PT interventions. It includes 18 items, and the parents rate each question on a 5-point scale. The items of the scale are usually analysed separately [16]. Higher scores indicate a greater degree of parental confidence.

#### 2.2.3. Parent–Child Relationship Measures

Assessment of the parent–child relationship was conducted using the New TK Diagnostic Test for Parent–Child Relationship [33]. This test evaluates parents’ attitudes and discipline from both parents’ and children’s perspectives. Elementary and middle school versions were used. The questionnaire contains 140 items for parents (70 items each for mothers and fathers) and 152 items for children, all of which are rated on a 4-point scale. A lower score indicates a worse parent–child relationship. Internal consistencies of the subscales in this study were adequate in 8 subscales (α = 0.72–0.85, Parent scale: Blame, Expectations, and Disagreement; Child scale: Expectations, Worry, Doting, Obedience, and Contradiction); however, Cronbach’s alpha values of other scales were below 0.7 (α = 0.15–0.68).

#### 2.2.4. Parents’ Statements Regarding Parent Training

At the end of session 6, facilitator NM asked the parents, ‘Please tell me what you think of PT, what has changed for you and your child, or what you would like to do in the future?’ The content of the statements obtained from the last PT session related to self-feedback on participation and awareness of changes in parents and children was analysed using the KJ method [34].

### 2.3. The Intervention

Parent training in this study was based on the PT programme of Iwasaka et al. [16] and Okuno et al. [15], following the programme of the Japan Parent Training Study Group (https://parent-training.jp/agreement.html, accessed on 1 November 2021). Okuno et al. [15] used the following contents: (1) ASD characteristics, observing, and understanding child behaviours; (2) how to focus on good behaviours and three categories of behaviours (appropriate, not-so-appropriate, inappropriate); (3) how to give clear instructions to their children, and how to not focus on the child’s inappropriate behaviours; (4) how to make and use a token table; (5) warnings and timeouts; and (6) how to teach a child control of emotions, and how to cooperate with the school. The adolescent version of this study added the following contents: adolescent features about DLSs, teaching parents the DLSs needed for independence in adulthood (e.g., hygiene, self-care, laundry, cooking, and money management), need for DLSs, decisions of DLSs to target (one to three target behaviours), and how to make DLS support items. Table 1 presents the detailed content of the programme. The programme consisted of six sessions in total (90 min each) and spanned two to three months, with sessions held every two weeks, followed by one final follow-up session three months later. PT was conducted in small groups of 2–4 participants. Parental attendance was 94.7%.

The first author (NM), a licensed clinical developmental psychologist who completed PT training in PT workshops held by the Japan Parent Training Study Group, facilitated the programme. The PT programme sessions were video recorded. The last author, HO, supervised programme completion and confirmed that the facilitator, NM, facilitated participants according to the PT protocol with the trainer checklist [16]. The checklist focuses on whether the facilitator did the following: (1) advised parents based on the ABC approach, (2) explained the three types of behaviours (appropriate, not-so-appropriate, inappropriate), (3) taught parents how to focus on and praise adolescents’ favourable behaviours, (4) explained the goals of the adolescents’ behaviour modification to parents and gave advice to parents, (5) set up consultations with parents and explained specific points of instruction, (6) explained to parents how to ignore inappropriate behaviours in their children, (7) explained how to work on homework, (8) gave appropriate feedback on homework, (9) understood adolescents’ characteristics, and (10) gave appropriate advice to parents who were unable to digest the session. The average protocol adherence rate was high (87.8%). Therefore, the current programme was considered adequately completed.

### 2.4. Date Analyses

None of the participants in the IT group dropped out of this study. One participant in the DTC group withdrew from participation after allocation and pre-investigation (Figure 1). Overall, 2 of the 24 participants were excluded from the analysis although they completed the entire session. One IT group participant did not return the questionnaire by the due date (two weeks) at Time 2, and one DTC group participant was found to have a child with an IQ above 100 on the WISC-IV administered after Time 2. Table 2 presents the children’s demographic variables. The labels and *n* represent the number of participants for each data element/analysis in all tables. Two participants with missing values were excluded from the analysis: one for the IT group regarding SRS-2 and one for the DTC group regarding the New TK Diagnostic Test for Parent–Child Relationship (Child).

Statistical analyses were performed using the IBM SPSS Statistics version 25. The analysis targeted changes in children’s behaviour, parenting, and the parent–child relationship. Group differences at pre-intervention were analysed using a *t*-test or Chi-square test, as appropriate. Analyses of covariance (ANCOVAs) were used to investigate changes in scores between Time 1 and Time 2. ANCOVA was not performed when any assumptions (e.g., homogeneity of the regression line slopes) were not met. The significance level was set at 5% for all analyses.

The KJ method [31] was used for the analysis of parents’ statements: statements were categorised by element and compared to each other using one statement on one card. To maintain validity, three researchers classified the content of the parents’ statements, of which two were graduate students and elementary school teachers and one was a clinical psychologist with clinical experience in adolescent care. None of these researchers was involved in the PT sessions.

## 3. Results

### 3.1. Children’s Changes

ANCOVA showed no significant improvement in the VABS-II DLS scores in the treatment group compared to the DTC group (Table 3). The VABS-II communication scores showed a significant effect of the intervention (partial *η*^2^ = 0.22). No significant effects of the intervention were detected on the CBCL and SRS-2 indices.

Regardless of the lack of improvement in the treatment group in terms of their VABS-II DLS scores, a portion (75%) of the IT group achieved individually tailored DLS behaviours after PT. Table 4 presents the individual profiles in terms of their acquisition of the DLSs.

### 3.2. Parents’ Changes

Table 5 shows the mean scores, standard deviations, and ANCOVA results for the PSI and BDI-II. No significant effects of the intervention were detected on the PSI and BDI-II scores.

Table 6 shows the results of the ANCOVA for each item of the CDQ. The fourth item on the CDQ, ‘Praise your adolescents more than once a day’, showed significant improvement after the intervention (partial *η*^2^ = 0.20). No significant effects of the intervention were detected for other CDQ items.

### 3.3. Parent–Child Relationship Changes

One child in the DTC group did not complete the questionnaire and was excluded from the analysis. The parents’ and children’s ANCOVA results for the New TK Diagnostic Test for Parent–Child Relationship are shown in Table 7 and Table 8, respectively. None of the children’s responses showed a significant effect of intervention. Regarding the parents’ responses, the IT group showed worse scores for the dissatisfaction measure than the DTC group at Time 2.

### 3.4. Analysis of Parents’ Statements Using the KJ Method

Parents’ statements were divided into five categories (Appendix A Table A1). Parents reported praising their children, ignoring dysfunctional behaviours, and using tokens (PT techniques). They detailed changes in their thoughts, perceptions, and emotional states as well as their and their children’s behaviours (changes after receiving PT). One participant emphasised the self-determination of children during adolescence (PT for adolescents). They also described the future challenges for adolescent children and themselves (future tasks).

## 4. Discussion

In this study, we conducted PT with the parents of adolescents with ASD and examined the effects of the intervention. Our results showed that 75% of the adolescents achieved the target DLSs after the intervention, although their DLS domain score for the VABS-II did not show significant improvement compared to the DTC group. The communication domain of the VABS-II and one item of the CDQ improved significantly. Regarding the New TK Diagnostic Test for the Parent–Child Relationship, there was a slight deterioration in the dissatisfaction score.

### 4.1. Children’s Changes

Nine (75%) adolescent children achieved the target DLS (Table 4). However, the general DLS measured using the VABS-II DLS domain did not show a significant increase. In contrast to the current training involving 12 weeks, a recent intervention programme improved DLSs in adolescents with ASD [35] over a course of more than 15 weeks. In addition, similar programmes [35,36] have prompted parents to teach and demonstrate skills to and discuss skills with their children. However, these two factors were not considered in this study. Therefore, PTs might require a longer duration of intervention and additional DLS training for the acquisition of DLSs in adolescents with ASD. Another possible factor was the participants’ prior experiences of the professional support for parenting or childcare. Thus, most of the participants (20/22) had received some of the support for parenting or childcare, which may affect parenting skills and knowledge about ASD characteristics in the parents. The lack of changes may have affected their prior experience of professional support. Most of the participants, both parents and children, had received intervention support. Parents were considered to have parenting skills for children with ASD. Therefore, we believe that the lack of change in the parents in this study was due to the fact that they already had parenting skills for children with ASD, and therefore the PT was less effective. Because DLSs improve the quality of daily living of individuals with ASD [5,35,36,37], investigation of these factors will be an important mission to develop training for adolescents with ASD.

An improvement was observed in the communication domain of VABS-II. Increased parent–child involvement through the intervention encouraged communication, which may lead to improved communication skills in children, as reported in previous studies [16,22]. Thus, PT interventions may have a positive effect on improving communication skills in adolescents.

### 4.2. Parents’ Changes

Regarding the CDQ, there was an improvement in Q4 (Do you praise your child once or more a day?), which is related to establishing skills to praise adolescents. It might indicate that parents learned about the necessity and effects of complimenting adolescent children, as in other Japanese PT studies [15,16]. There were no significant improvements in the other CDQ items. Compared to Okuno et al. [38], who conducted PT in younger children, improvements in CDQ scores were limited in this study. It is possible that a younger age in children is associated with greater improvement. No significant improvements were observed in the PSI and BDI-II scores. The lack of change in these scores might be due to the floor effect because the level of distress among the parents was low in this study.

### 4.3. Parent–Child Relationship Changes

Regarding the New TK Diagnostic Test for Parent–Child Relationship, there was a slight increase in the dissatisfaction score of parents in the IT group. However, careful interpretation is required for this result, because the internal consistency of the subscale was low. Although some characteristics of the PT, such as spending time with children, may affect the results, the relationship should be examined other reliable measures in future studies.

### 4.4. Limitations

This study has some limitations in terms of sampling and research design. Because we recruited families from a university hospital, the representativeness of the sample and the generalisability of the results are under scrutiny. Additionally, the sample size was too small to yield a definitive conclusion regarding the effectiveness of the intervention. Prior sample size calculation should be required to obtain more reliable conclusions. Regarding the research design aspect, the intervention period might have been shorter than that in some previous studies. Additionally, we did not prepare a placebo condition and measures that did not involve subjective evaluations by the trainees themselves (i.e., parents). Future research will provide a definitive conclusion on the effect of training by considering these factors in the design. Finally, we did not perform a follow-up survey and could not track the families’ conditions after this training.

### 4.5. Implications for Research and Practice

Support for parents of adolescents with ASD is insufficient in Japan. Thus, this study provides a significant contribution to the field in the Japanese context. However, clear evidence must be provided to establish a DLS-specific PT for parents of adolescents with ASD. Moreover, long-term intervention may facilitate the establishment of DLSs more effectively. To acquire DLSs, we believe that it is necessary to provide support to parents and children with ASD so that they can practice such skills at home. In addition, it is also helpful to include content that was designated to reduce parent–child conflicts, depression, and stress. Conduction of the intervention by a therapist for both parents and children may increase its effectiveness, as in the case of a recent intervention programme by Duncan et al. [39]. Developing an evidence-based strategy to improve DLSs in adolescents is necessary for future research and practice to enhance the social functioning of adolescents with ASD in the community.

## 5. Conclusions

In this study, we conducted a DLS-specific PT for parents of adolescents with ASD and examined its effectiveness. Improvements were observed in the VABS-II communication domain and the confidence of parents in praising after the intervention. These results indicate that the PT programme was effective for some adaptive behaviours. While our study did not confirm the effectiveness of PT in general DLS, the findings suggest the potential utility of PT in adolescent children with ASD. Further research is required to establish the effectiveness of DLS interventions in supporting children with ASD.

## Figures and Tables

**Figure 1 ijerph-19-02363-f001:**
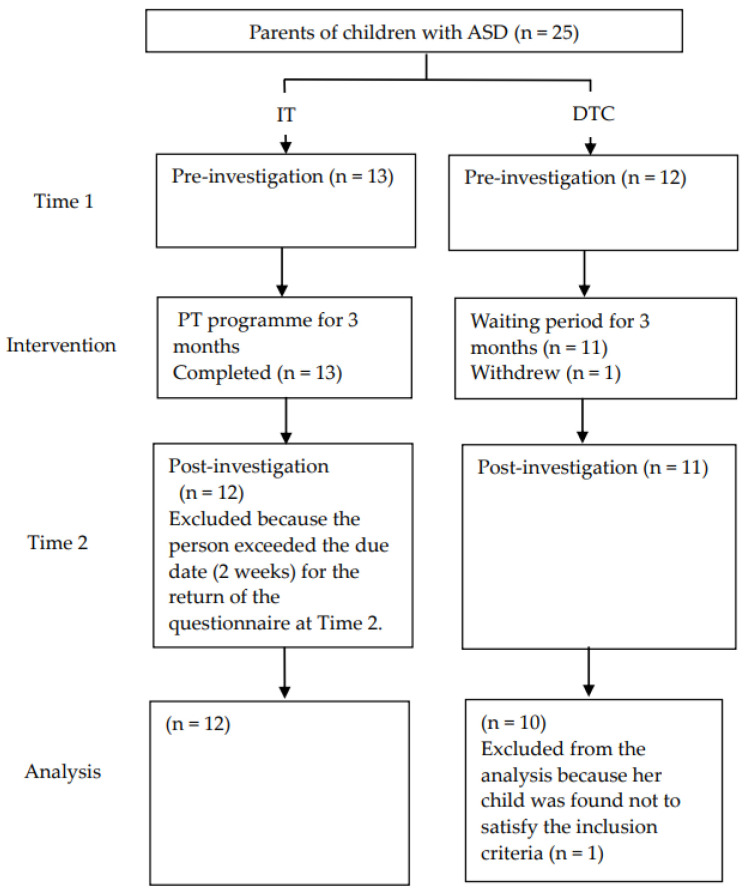
The flow of the participants in this study.

**Table 1 ijerph-19-02363-t001:** Contents of the PT sessions.

Session	Contents	Homework
〈Session 1〉 Behavioural observation and understanding of ASD characteristics	Self-introduction ASD characteristics, adolescent features About types of DLSs (e.g., hygiene, self-care, laundry, cooking, and money management) Observing and understanding child behaviours	Fill in the observation sheet: ‘child behaviour, parent’s response, and child’s reaction’
〈Session 2〉 Focus on good behaviours	How to focus on good behaviours Three categories of behaviours: appropriate, not-so-appropriate, and inappropriate	Fill in the observation sheet: ‘How the parent praised the child’s behaviours and dividing children’s behaviours into three categories’
〈Session 3〉 Instructions that are easy for children to understand	How to give clear instructions to their children How to not focus on child’s inappropriate behaviours	Fill in the observation sheet: ‘The child’s behaviours when the parent gives instructions to the child and the subsequent behaviours of the child’.
〈Session 4〉 Token economy	Need for DLSs Decisions of which DLSs to target (1–3 target behaviours)	Assess the child’s current DLSs and decide which DLSs the child will practise
	How to make and use a token table	
〈Session 5〉 DLS support items Limit setting	How to make DLS support items Warnings and timeouts	Teaching DLSs to the child Using limitation skills
〈Session 6〉 Cooperation with school Summary	How to teach control of emotions to a child How to cooperate with the school Summary of and reflection on DLSs	
〈Follow-up〉 Conducted three months after the end of Session 6	Check the status of the child after PT implementation.	

**Table 2 ijerph-19-02363-t002:** Demographic characteristics of the immediate-treatment (IT) and delayed-treatment control groups (DTC).

Variable			Group		*T*	*χ* ^2^	*p*
			IT (*n* = 12)	DTC (*n* = 10)			
Children							
Age		Mean (SD)	12.42 (1.61)	11.50 (1.43)	1.40		0.18
Gender	Male	*n*	11	9		0.02	0.89
	Female	*n*	1	1			
WISC-IV	Full-scale IQ	Mean (SD)	81.33 (9.02)	85.20 (10.26)	−0.94		0.36
Type of schooling	Regular class	*n*	1	3		0.19	0.23
	Special education class	*n*	11	7			
Experience of professional support for parenting or childcare	Yes	*n*	10	10		0.18	0.29
	None	*n*	2	0			
Sibling	None	*n*	6	2		2.84	0.24
	One	*n*	5	5			
	Two or more	*n*	1	3			
Parents							
Age		Mean (SD)	44.58 (4.76)	43.70 (4.19)	0.46		0.65
University degree	Yes	*n*	5	3		0.64	0.73
	No	*n*	5	4			
	Information not available	*n*	2	3			
Marital status	Single	*n*	0	1		1.26	0.26
	Married	*n*	12	9			

Note. SD: standard deviation.

**Table 3 ijerph-19-02363-t003:** Mean scores, standard deviations, and ANCOVA results for indicators measuring adolescents’ changes.

Measure		Group	*n*	Time 1		Time 2		*F*	*p*	Partial *η*^2^
				Mean	SD	Mean	SD			
VABS-II	Composite	IT	12	49.80	7.76	56.40	5.95	—		
		DTC	10	58.80	13.41	60.10	11.88			
	**Communication**	IT	12	46.92	10.94	55.75	9.54	5.43	0.03	0.22
		DTC	10	59.40	15.44	58.00	13.93			
	DLSs	IT	12	60.30	11.61	68.30	11.43	0.63	0.44	0.03
		DTC	10	69.70	10.61	72.60	9.89			
	Socialisation	IT	12	55.75	10.42	62.25	4.51	—		
		DTC	10	61.90	14.76	63.60	13.60			
CBCL	Internalisation T score	IT	12	63.92	4.54	61.00	7.22	1.14	0.30	0.06
		DTC	10	64.20	6.95	63.80	8.68			
	Externalisation T score	IT	12	54.58	7.79	55.58	9.16	2.65	0.12	0.12
		DTC	10	62.30	8.74	59.10	10.47			
SRS-2	SCI T score	IT	11	68.27	5.88	65.64	5.03	—		
		DTC	10	73.00	10.68	71.70	10.28			
	RRB T score	IT	11	70.73	10.33	72.09	11.65	0.79	0.38	0.04
		DTC	10	69.70	14.50	68.00	15.25			
	Social awareness	IT	11	56.27	6.89	59.45	7.71	1.14	0.30	0.06
		DTC	10	64.00	7.44	60.80	9.61			
	Social cognition	IT	11	70.64	9.48	68.27	9.00	1.67	0.21	0.09
		DTC	10	73.90	10.31	73.90	8.91			
	Communication	IT	11	68.55	5.74	65.36	7.94	1.09	0.31	0.06
		DTC	10	73.10	10.63	71.90	10.52			
	Social motivation	IT	11	61.27	14.72	58.27	14.60	0.22	0.65	0.01
		DTC	10	65.00	13.69	62.00	6.34			
	Restricted interest and repetitive behaviour	IT	11	71.18	9.86	72.09	11.65	0.73	0.40	0.04
		DTC	10	70.00	14.32	68.00	15.25			

Note. A hyphen indicates that ANCOVA was not available where linearity was not satisfied. Bolded label measures in each table indicate that we observed significant effects.

**Table 4 ijerph-19-02363-t004:** Achieved target DLSs for each child.

Child’s Number	DLSs Determined as a Goal at Session 4	Achievement Classification of DLSs (Mother’s Report)	Other DLSs Conducted
1	After bathing, drying own hair Preparation of school belongings for the next day	1 1	
2	After returning home, cleaning up own belongings	3	Preparing what to bring to school the next day
3	Reducing nail biting	2	
4	Preparation from getting up to going to school	3	Bathing at eight o’clock Sleeping alone Washing the water bottle
5	Cleaning up after meals	2	
6	Taking the rubbish out to the rubbish dump in the morning	1	
7	Cleaning up their room Putting their clothes in the closet	2 2	Wiping the table at mealtime Helping with housework and school preparation for the next day
8	Washing their uniform shirt Folding laundry Closing the curtains	2 2 2	
9	Pouring tea into a water bottle in the morning	1	Being able to go to the dentist without an attendant
10	Morning preparation	3	Help with cleaning
11	Managing their medication by themselves	2	Preparation before going to bed and meal preparation
12	Cleaning the bath (1–2 times a week)	2	

Note. The children’s achievement of DLSs between sessions 5 and 6 was evaluated by their parents in regard to each task that was set after parent–child discussions as part of the session 4 assignments. Parents rated their child’s behaviours in regard to DLSs on a 3-point scale. Achievement classification: 1 = Independent, 2 = Likely to be independent (partially prompted, increase in practice), 3 = Not continuing or not practicing 75.0% achieved the target behaviours (DLSs were classified as 1 or 2; 9 out of 12).

**Table 5 ijerph-19-02363-t005:** Mean scores, standard deviations, and ANCOVA results for indicators measuring parents’ changes.

Measure		Group	*n*	Time 1		Time 2		*F*	*p*	Partial *η*^2^
				Mean	SD	Mean	SD			
PSI Parenting Stress Index	Parent Total	IT	12	111.00	12.63	110.41	16.69	0.01	0.91	<0.01
		DTC	10	115.30	24.03	113.00	19.91			
	Child Total	IT	12	101.42	14.82	102.08	12.43	1.80	0.20	0.09
		DTC	10	116.60	15.60	108.90	20.54			
BDI-II	Total score	IT	12	11.17	6.59	12.25	8.31	3.86	0.06	0.17
		DTC	10	13.00	8.76	10.20	7.36			

**Table 6 ijerph-19-02363-t006:** Mean scores, standard deviations, and ANCOVA results for the Confidence Degree Questionnaire for families (CDQ).

Measure		Group	*n*	Time 1		Time 2		*F*	*p*	Partial *η*^2^
				Mean	SD	Mean	SD			
Q 1	Do you watch your child’s growth without becoming impatient?	IT	12	2.83	0.90	3.17	0.69	0.90	0.35	0.05
		DTC	10	2.80	0.98	2.90	0.94			
Q 2	Do you accept your child’s diagnosis of ASD?	IT	12	4.67	0.47	4.33	0.85	0.35	0.56	0.02
		DTC	10	3.40	1.50	4.10	0.70			
Q 3	Do you let your child do what he/she can do by him/herself?	IT	12	3.58	0.76	3.42	0.95	0.16	0.70	0.01
		DTC	10	3.70	0.90	3.60	0.66			
**Q 4**	**Do you praise your child once or more a day?**	IT	12	3.08	1.26	3.75	1.23	4.70	0.04	0.20
		DTC	10	3.10	1.04	2.90	1.22			
Q 5	Do you prepare a place where your child can relax?	IT	12	3.67	1.11	3.50	1.04	1.45	0.24	0.07
		DTC	10	3.00	0.89	3.70	1.00			
Q 6	Do you help your child to make friends?	IT	12	3.08	0.95	3.08	1.11	0.37	0.55	0.02
		DTC	10	3.00	1.10	2.80	0.87			
Q 7	Can you cope with your child’s inappropriate behaviour?	IT	12	2.92	1.11	3.33	0.62	0.76	0.39	0.04
		DTC	10	3.30	0.90	3.10	0.70			
Q 8	Do you communicate adequately with the school about your child’s problems in school?	IT	12	3.75	0.83	3.58	0.76	<0.01	0.96	<0.01
		DTC	10	3.60	0.80	3.50	0.67			
Q 9	Do you blame yourself less for having a child with ASD?	IT	12	3.25	0.72	3.33	0.85	0.09	0.77	<0.01
		DTC	10	3.20	0.87	3.20	0.98			
Q 10	Are you less worried about your child?	IT	12	2.75	0.72	2.75	0.83	0.06	0.80	<0.01
		DTC	10	2.60	1.02	2.70	1.10			
Q 11	Do you spend time on your own health or enjoyment?	IT	12	3.00	0.91	3.25	0.83	0.02	0.88	<0.01
		DTC	10	3.70	1.00	3.60	1.20			
Q 12	Do you quarrel less with your family due to your child’s behaviour?	IT	12	3.08	0.86	2.75	1.09	3.04	0.10	0.15
		DTC	10	2.80	0.98	3.20	0.75			
Q 13	Do you ask your family members to assist your child?	IT	12	3.25	1.01	2.92	1.04	0.02	0.90	<0.01
		DTC	10	2.50	1.02	2.60	1.11			
Q 14	Do you consult your family or friends about your troubles and not worry by yourself?	IT	12	3.92	1.11	4.00	0.82	―		
		DTC	10	3.40	1.20	3.50	1.28			
Q 15	Do you share your feelings with families who have children with a similar problem?	IT	12	3.42	1.26	3.58	1.04	<0.01	0.99	<0.01
		DTC	10	3.30	1.19	3.50	1.12			
Q 16	Do you utilise medical facilities and school and consultative organisations if required?	IT	12	3.92	0.86	3.92	0.76	1.39	0.26	0.07
		DTC	10	4.00	0.77	4.30	0.90			
Q 17	Do you understand your child’s behaviours and ideas/feelings/thoughts?	IT	12	3.00	1.15	3.58	0.98	3.96	0.06	0.17
		DTC	10	3.20	0.98	2.80	1.07			
Q 18	Do you feel happy being with your child?	IT	12	3.83	1.34	3.92	1.04	0.25	0.62	0.01
		DTC	10	3.60	1.02	3.60	1.11			

Note. A hyphen indicates that ANCOVA was not available where linearity was not satisfied. Bolded label measures in each table represent that we observed significant effects.

**Table 7 ijerph-19-02363-t007:** Mean scores, standard deviation, and ANCOVA results for the New TK Diagnostic Test for Parent–Child Relationship (Child).

Measure	Group	*n*	Time 1		Time 2		*F*	*p*	Partial *η*^2^
			Mean	SD	Mean	SD			
Dissatisfaction	IT	12	25.83	3.05	24.58	5.13	0.75	0.39	0.04
	DTC	9	24.56	2.67	24.67	2.78			
Blame	IT	12	24.67	3.59	23.75	4.81	0.05	0.83	<0.01
	DTC	9	27.00	1.76	24.89	3.06			
Strictness	IT	12	24.83	4.71	23.25	5.51	0.15	0.70	0.01
	DTC	9	25.56	2.67	23.44	4.56			
Expectations	IT	12	25.58	4.96	24.50	4.82	0.01	0.94	<0.01
	DTC	9	24.33	5.75	23.67	5.79			
Interference	IT	12	22.17	4.74	21.89	5.22	<0.01	0.99	<0.01
	DTC	9	23.00	4.24	22.44	4.93			
Worry	IT	12	21.75	5.60	22.00	4.73	0.71	0.41	0.04
	DTC	9	21.22	4.66	23.11	4.89			
Doting	IT	12	18.83	6.00	20.83	5.96	0.52	0.48	0.03
	DTC	9	21.56	4.22	23.67	4.24			
Obedience	IT	12	22.17	5.34	23.08	4.94	0.12	0.73	0.01
	DTC	9	23.00	2.11	24.22	4.32			
Contradiction	IT	12	22.58	4.86	21.75	5.36	0.30	0.59	0.02
	DTC	9	25.56	3.13	24.56	3.25			

**Table 8 ijerph-19-02363-t008:** Mean score, standard deviation, and ANCOVA results for the New TK Diagnostic Test for Parent–Child Relationship (Mother).

Measure	Group	*n*	Time 1		Time 2		*F*	*p*	Partial *η*^2^
			Mean	SD	Mean	SD			
Dissatisfaction	IT	12	24.50	1.26	23.08	2.11	6.41	0.02	0.25
	DTC	10	21.60	2.11	21.80	3.40			
Blame	IT	12	23.00	3.61	23.33	3.92	0.26	0.62	0.01
	DTC	10	21.90	4.01	21.90	4.25			
Strictness	IT	12	23.25	2.35	23.42	3.20	0.40	0.53	0.02
	DTC	10	22.30	2.72	22.10	2.91			
Expectations	IT	12	24.33	3.25	24.83	4.55	—		
	DTC	10	24.20	3.79	24.50	3.34			
Interference	IT	12	20.17	2.48	21.33	4.08	0.44	0.51	0.02
	DTC	10	19.70	2.79	20.10	3.04			
Worry	IT	12	23.92	3.40	22.83	3.76	0.18	0.67	0.01
	DTC	10	22.50	2.54	22.30	3.40			
Doting	IT	12	24.75	3.37	24.17	3.59	0.39	0.54	0.02
	DTC	10	23.70	1.27	23.80	2.30			
Obedience	IT	12	24.08	3.12	23.08	3.99	1.34	0.26	0.07
	DTC	10	23.80	2.27	24.00	2.83			
Contradiction	IT	12	24.42	2.10	25.00	4.07	—		
	DTC	10	23.30	2.15	23.70	1.64			
Disagreement	IT	12	23.67	3.22	25.00	3.08	0.37	0.55	0.02
	DTC	10	22.30	5.33	24.70	4.37			

Note. A hyphen indicates that ANCOVA was not available where linearity was not satisfied.

## Data Availability

The data of this study are not available because sharing the data with third parties not listed in the original protocol was not approved by the ethics committee. The data are not publicly available because of ethical restrictions.

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
