# Peer review of "Effectiveness of a Parent Training Programme for Parents of Adolescents with Autism Spectrum Disorders: Aiming to Improve Daily Living Skills"

_ijerph, 2022, doi:10.3390/ijerph19042363_

Round 1
Reviewer 1 Report
I always read manuscripts about improving the quality of life of patients with ASD with great hope and interest. ASD is a problem for many families and every opportunity to improve their situation is important and necessary.
The manuscript is clearly prepared. I like the well-described limitation section, which shows the high scientific level of the authors. The study is properly planned and carried out. I have no major comments, apart from a few comments:
- Some sections are justified, some are not.
- Section 3.4 must have line spacing corrected.
- Line 375 must have citation numbers only.
Author Response
Manuscript ID: ijerph-1519558
Type of manuscript: Article
Title: Effectiveness of a Parent Training Programme for Parents of Adolescents with Autism Spectrum Disorders: Aiming to Improve Daily Living Skills
We appreciate the reviewers’ supportive and constructive comments for improving our manuscript.
Response to the Reviewer 1
General comment:
I always read manuscripts about improving the quality of life of patients with ASD with great hope and interest. ASD is a problem for many families and every opportunity to improve their situation is important and necessary. The manuscript is clearly prepared. I like the well-described limitation section, which shows the high scientific level of the authors. The study is properly planned and carried out. I have no major comments,
Response:
We appreciate the reviewer’s positive feedback on our manuscript.
Comment 1:
Some sections are justified, some are not.
Response 1:
We have reviewed the entire manuscript and revised the format.
Comment 2:
Section 3.4 must have line spacing corrected.
Response 2:
The line spacing was corrected.
Comment 3:
Line 375 must have citation numbers only.
Response 3:
We have corrected the citation style according to the author’s guideline.

Reviewer 2 Report
This manuscript describes a parenting intervention designed to enhance the daily living skills of adolescents with autism spectrum disorder. The work offers clinical and research implications. I have several queries / suggestions for improvement but think the work has merit.
1. The design of the study needs clarification, particularly with regard to sample size. This is small – was this study intended to be a pilot evaluation? If so, this should be clearly stated. If not, how was the sample size decided upon? Was any consideration given to statistical power?
2. A related point is that more detail is required about the sample and recruitment. Did Osaka University Hospital provide care to all families of ASD children in the region, or only those with severe difficulties? Was recruitment through consecutive referrals / attendances at the hospital or was it more targeted? How many families were invited, ie, what was the percentage agreement rate amongst those invited to participate?
3. Presumably one reason parenting interventions usually focus on younger children is that by the time children reach adolescence, parents would (ideally) have attended a parenting intervention already. Consistent with this, Table 2 suggests that 20/22 participants had prior professional support for ‘parenting or childcare’. Could the Authors clarify if this means most parents had previously had a parenting skills intervention of some kind? This would reduce the potential for effects in the current study and may help explain the null results.
4. Please provide alpha values (internal consistency) for each of the measures used in the current sample.
5. CDQ measure and analysis. Is it usual for the CDQ items to be analysed separately (as done here) or would these usually be treated as a composite score?
Author Response
Manuscript ID: ijerph-1519558
Type of manuscript: Article
Title: Effectiveness of a Parent Training Programme for Parents of Adolescents with Autism Spectrum Disorders: Aiming to Improve Daily Living Skills
We appreciate the reviewers’ supportive and constructive comments for improving our manuscript.
Response to the Reviewer 2 
This manuscript describes a parenting intervention designed to enhance the daily living skills of adolescents with autism spectrum disorder. The work offers clinical and research implications. I have several queries / suggestions for improvement but think the work has merit.
Response:
We appreciate the reviewer’s thoughtful comment on our manuscript.
Comment 1a:
The design of the study needs clarification, particularly with regard to sample size. This is small – was this study intended to be a pilot evaluation? If so, this should be clearly stated. If not, how was the sample size decided upon? Was any consideration given to statistical power?
Response 1a:
Although this study was not a pilot evaluation, its sample size was limited. Unfortunately, we did not estimate the required sample size prior to recruitment. Based on a previous Japanese study (Okuno et al.,2011), we stopped recruiting once we achieved similar numbers in each group. The sample size was 25 parents of children with ASD who declared their participation. We have described further details about the recruitment of the family in the Methods and Limitations section.
Post hoc power analysis was not conducted because it does not add meaningful information to the readers (Hoenig & Heisey, 2001).
In Methods, Line 105
We recruited parents of patients seen at Osaka University Hospital during the recruitment period (August-November 2018). The recruitment method was as follows: when parents who met the criteria came to the hospital, their doctors or psychologists gave them a Parent Training leaflet and explained about PT. Based on a previous Japanese study [22], we stopped recruiting once we achieved similar numbers in each group, and the sample size was 25 parents of the children with ASD who declared their participation.
Citation
Okuno, H.; Nagai, T.; Sakai, S.; Mohri, I.; Yamamoto, T.; Yoshizaki, A.; Kato, K.; Tachibana, M.; Iwasaka, H.; Taniike, M. Effectiveness of Modified Parent Training for Mothers of Children with Pervasive Developmental Disorder on Parents Confidence and Children’s Behavior. Brain Dev. 2011, 33 (2), 152–160.
Comment 1b:
Line 454 In addition, the sample size was small to yield a definitive conclusion for the effectiveness of the intervention. Prior sample size calculation should be required to obtain more reliable conclusions.
Response 1b:
The reviewer highlighted an important issue in the design of this study. Based on the reviewer’s suggestion, we clarified the need for prior sample size calculation in the limitations section.
In Limitation, Line 443
Additionally, the sample size was too small to yield a definitive conclusion regarding the effectiveness of the intervention. Prior sample size calculations are required to obtain reliable conclusions.
Comment 2:
A related point is that more detail is required about the sample and recruitment. Did Osaka University Hospital provide care to all families of ASD children in the region, or only those with severe difficulties? Was recruitment through consecutive referrals / attendances at the hospital or was it more targeted? How many families were invited, ie, what was the percentage agreement rate amongst those invited to participate?
Response 2:
Osaka University Hospital is a tertiary care institution (at the time of the study), and there was a variety in the degree of symptoms in patients. We consecutively recruited parents who met the inclusion criteria among those who visited the hospital during the recruitment period (August to November 2018). Unfortunately, the total number of parents referred to the study and agreement rate was unavailable. This limited the representativeness of the sample. We have added the following sentence in the limitations section:
In Limitation line 441
Because we recruited the families from a university hospital, the representativeness of the sample and generalisability of the results is under scrutiny.
Comment 3:
Presumably one reason parenting interventions usually focus on younger children is that by the time children reach adolescence, parents would (ideally) have attended a parenting intervention already. Consistent with this, Table 2 suggests that 20/22 participants had prior professional support for ‘parenting or childcare’. Could the Authors clarify if this means most parents had previously had a parenting skills intervention of some kind? This would reduce the potential for effects in the current study and may help explain the null results.
Response 3:
The reviewer highlighted a very important point for discussion. Twenty parents received one of the following professional support: parent training, counselling, applied behaviour analysis, occupational therapy, and speech therapy. Thus, most participants had received some support for parenting or childcare, which may have affected parenting skills and knowledge about ASD characteristics. As the reviewer suggested, the lack of change in this study may be due to their prior experience with professional support. We have added this information in the Methods section and discussed it in the Discussion section.
In Methods
- Line116: Twenty parents were had experience in receiving support about parenting or childcare (i.e., parent training, counselling, applied behaviour analysis), occupational therapy, and speech therapy).
In Discussion
- Line 379: Another possible factor was the participants’ prior experience with professional support for parenting or childcare. Thus, most of the participants (20/22) had received support for parenting or childcare, which may have affected their parenting skills and knowledge about ASD characteristics. This lack of change may have affected their prior experience of professional support.
Comment 4:
Please provide alpha values (internal consistency) for each of the measures used in the current sample.
Response 4:
The alpha values (internal consistency) for the relevant ratings have been added. Most scales showed adequate internal consistency; however, some showed low alpha values. We have added this information in the Methods section and revised the Discussion section.
In Methods
- Line 163: Internal consistencies of the scales in this study were adequate (Cronbach’s α = 0.68; internalising scale, α = 0.86 for externalising scale).
- Line 175: Internal consistencies of the subscales in this study were acceptable to a high of 0.61-0.85, except for social awareness (Cronbach’s α = 0.18).
- Line 184: Internal consistencies of the subscales were high (Cronbach’s α = 0.81 for parent total and α = 0.85 for child total).
- Line 192: The internal consistency was high in this study (Cronbach’s α = 0.89).
- Line 211: Internal consistencies of the subscales in this study were adequate in 8 subscales (α = 0.72-0.85, Parent scale: Blame, Expectations, and Disagreement; Child scale: Expectations, Worry, Doting, Obedience, and contradiction); however, Cronbach’s alpha of other scales were below 0.7 (α = 0.15-0.68).
In Discussion
- Line 424:3. Parent-child relationship changes
Regarding the New TK Diagnostic Test for Parent-Child Relationship, there was a slight increase in the dissatisfaction score of parents in the IT group. However, careful interpretation is required for this result, because the internal consistency of the subscale was low. Although some characteristics of the PT, such as spending time with children, may affect the results, the relationship should be examined other reliable measures in future studies.
Comment 5:
CDQ measure and analysis. Is it usual for the CDQ items to be analysed separately (as done here) or would these usually be treated as a composite score?
Response 5:
The CDQ has not been developed as a measure for calculating composite scores. Therefore, it is usual for the CDQ items to be analysed separately. The PT program in this study was structured based on the Japan Association of Parent Training program, and the association recommended the CDQ to evaluate the effectiveness of PT interventions.
We have clarified the following in the Methods section: The Japan Association of Parent Training has been promoting the CDQ to evaluate the effectiveness of PT interventions.
In Methods, Line 214
The Japan Association of Parent Training recommends the CDQ for evaluating the effectiveness of PT interventions. It includes 18 items, and the parents rate each question on a 5-point scale. The items of the scale are usually analysed separately [16].
- Iwasaka, H.; Shimizu, T.; Iida, J.; Kawabata, Y.; Chikaike, M.; Onishi, T., et al. Efficacy of a Parenting Program as Attention/Hyperactivity Disorder (AD/HD) Therapy (in Japanese). Jido Seinen Seishin Igaku to Sonogr. Kinsetsu Ryoiki 2002, 43, 483–497.
- Okuno,H.;Nagai, T.; Mohri, I.; Yoshizaki,A.;Yamamoto, T.;Sakai, S.;Iwasaka, H.;Taniike, M. Effectiveness of a Modified Parent Training of Smaller Group and Shorter Schedules for Children with Pervasive Developmental Disorders. No To Hattatsu.2013;45:26-32.
Other revisions:
A professional English editing service has proofread the entire manuscript.

Reviewer 3 Report
This is a well-written manuscript, with a well-conducted study on the effects of parental training. The analyses are appropriate, and very thorough.
Author Response
Manuscript ID: ijerph-1519558
Type of manuscript: Article
Title: Effectiveness of a Parent Training Programme for Parents of Adolescents with Autism Spectrum Disorders: Aiming to Improve Daily Living Skills
We appreciate the reviewers’ supportive and constructive comments for improving our manuscript.
Response to the Reviewer 3 
Comment:
This is a well-written manuscript, with a well-conducted study on the effects of parental training. The analyses are appropriate, and very thorough.
Response:
We appreciate the positive feedback from the reviewer. A professional English language editing service has proofread the entire manuscript.

Reviewer 4 Report
The article addresses a very important issue of the relationship between parents and adolescent ASDs. The structure of the article in terms of content and research does not require improvement. The only remark is the relatively small size of the group in relation to the quantitative research.
Author Response
Manuscript ID: ijerph-1519558
Type of manuscript: Article
Title: Effectiveness of a Parent Training Programme for Parents of Adolescents with Autism Spectrum Disorders: Aiming to Improve Daily Living Skills
We appreciate the reviewers’ supportive and constructive comments for improving our manuscript.
Response to the Reviewer 4
Comment:
The article addresses a very important issue of the relationship between parents and adolescent ASDs. The structure of the article in terms of content and research does not require improvement. The only remark is the relatively small size of the group in relation to the quantitative research.
Response:
We appreciate the positive feedback from the reviewer. A professional English language editing service has proofread the entire manuscript.
